# Carborane-Based Analog of Rev-5901 Attenuates Growth of Colon Carcinoma In Vivo

**DOI:** 10.3390/molecules27144503

**Published:** 2022-07-14

**Authors:** Svetlana Paskaš, Blagoje Murganić, Robert Kuhnert, Evamarie Hey-Hawkins, Sanja Mijatović, Danijela Maksimović-Ivanić

**Affiliations:** 1Department of Immunology, Institute for Biological Research “SinišaStanković”, Belgrade University, 11060 Belgrade, Serbia; svetlana.paskas@ibiss.bg.ac.rs (S.P.); blagojemurganic@gmail.com (B.M.); sanjamama@ibiss.bg.ac.rs (S.M.); 2Institute of Inorganic Chemistry, Faculty of Chemistry and Mineralogy, Leipzig University, 04103 Leipzig, Germany; robertkuhnert@googlemail.com (R.K.); hey@uni-leipzig.de (E.H.-H.)

**Keywords:** 5-lipoxygenase, Rev-5901, colorectal carcinoma, CT26CL25, carboranes

## Abstract

Lipoxygenases convert polyunsaturated fatty acids into biologically active metabolites such as inflammatory mediators—prostaglandins and leukotrienes. The inhibition of lipoxygenases is increasingly employed in the treatment of cancer. We evaluated the anticancer potential of two novel 5-lipoxygenase inhibitors, named CarbZDNaph and CarbZDChin, which are analogues of the commercially available inhibitor Rev-5901. The in vitro segment of this study was conducted on a mouse colorectal carcinoma cell line—CT26CL25. For an in vivo model, we induced tumors in BALB/c mice by the implantation of CT26CL25 cells, and we treated the animals with potential inhibitors. A 48 h treatment resulted in diminished cell viability. Calculated IC_50_ values (half-maximal inhibitory concentrations) were 25 μM, 15 μM and 30 μM for CarbZDNaph, CarbZDChin and Rev-5901, respectively. The detailed analysis of mechanism revealed an induction of caspase-dependent apoptosis and autophagy. In the presence of chloroquine, an autophagy inhibitor, we observed an increased mortality of cells, implying a cytoprotective role of autophagy. Our in vivo experiment reports tumor growth attenuation in animals treated with CarbZDChin. Compounds CarbZDNaph and Rev-5901 lacked an in vivo efficacy. The results presented in this study display a strong effect of compound CarbZDChin on malignant cell growth. Having in mind the important role of inflammation in cancer development, these results have a significant impact and are worthy of further evaluation.

## 1. Introduction

Lipoxygenases are a class of enzymes that convert arachidonic acid into biologically active metabolites which act as autocrine and paracrine mediators [1]. Arachidonic acid and other polyunsaturated fatty acids can be converted by lipoxygenases into different signaling molecules such as hydroperoxy fatty acids, leukotrienes, resolvins, and lipoxins [2,3].

A key enzyme in the metabolism of arachidonic acid to leukotrienes is 5-lipoxygenase (5-LO). 5-LO is a soluble monomeric enzyme containing a non-heme iron, which catalyzes the oxidation of arachidonic acid to 5-HPETE and its further dehydration to leukotriene A4 (LTA4) [4]. This enzyme interacts with a protein located on the nuclear membrane called 5-lipoxygenase activating protein (FLAP). FLAP promotes the synthesis of LTA4 by “presenting” arachidonic acid to 5-LO [5].

Considerable evidence has accumulated verifying the pathophysiological role of the 5-lipoxygenase pathway in the development of allergic diseases such as asthma [6], inflammatory disorders such as rheumatoid arthritis and cardiovascular disease [7], and in tumorigenesis. 5-LO is found to be overexpressed in different human cancers such as prostate, pancreas, colon, urinary bladder, testis and renal carcinoma [8,9,10,11,12,13]. Several studies suggest that there is a link between 5-LO and carcinogenesis in humans and animals and that the inhibition of 5-LO has an anticarcinogenic effect in different tumor types [14,15,16].

The development of 5-LO inhibitors plays an important role in the therapy of the above-mentioned diseases. These compounds can be classified as (i) redox inhibitors: zileuton and BWA4C, acting by reduction of the ferric iron to the ferrous form, (ii) non-redox inhibitors: Rev-5901, AA-861 and CJ-13,610, competing with an arachidonic acid for binding to 5-LO, (iii) antagonists of leukotriene receptors: montelukast, zafirlukast and (iv) FLAP- inhibitors: MK-889, DG-031, acting by inhibiting FLAP [17,18].

In this study, we have investigated carborane-based analogs of Rev-5901. It has been previously shown that boron cluster-containing bioactive molecules are metabolically stable and non-toxic [19]. There are several modes of action of Rev-5901 described in the literature: inducing apoptosis by activation of caspases [20], cellular morphological changes resulting in a more differentiated phenotype [21], and cell growth inhibition [22]. Compounds **1** and **2** are naphthalene and quinoline modifications of ZD-2138 and Rev-5901 (CarbZDNaph and CarbZDChin, respectively), acting as FLAP inhibitors (Figure 1) [23].

Many studies presenting 5-LO inhibitors account for the limited selectivity of these compounds as well as possible off-targets and side effects. It has been shown that the common 5-LO inhibitors, such as AA-861, Rev-5901 and CJ-13,610 can reduce the viability of different cell lines independent of the suppression of 5-LO product formation [24]. In addition, well-established 5-LO negative tumor cell lines exhibited a higher susceptibility toward the 5-LO inhibitors than their morphologically related 5-LO expression counterparts [25]. These studies imply that the cytotoxic and chemo-preventive effects of these inhibitors in cell culture assays may derive from molecular mechanisms other than the suppression of leukotriene biosynthesis, and there is a need for the reassessment of their activity [26].

The in vitro model system used here is the murine colon carcinoma cell line CT26CL25, which is known to express the 5-lipoxygenase enzyme [27]. This study aims to gain insights into the mechanism of two Rev-5901 analogs, compounds **1** and **2** in colorectal carcinoma.

## 2. Results

### 2.1. Compounds **1** and **2** Induce Apoptosis, Activate Caspases and Cytoprotective Autophagy

In our previous publication, we demonstrated that 48 h treatment of CT26CL25 cells with compounds **1** and **2** resulted in diminished cell viability, and the IC_50_ values we used in this study were 25 μM, 15 μM, and 30 μM for **1**, **2** and Rev-5901, respectively [23].

Double staining with Annexin V/propidium iodide did not reveal the significant presence of apoptotic cells in the control and Rev-5901-treated CT26CL25 (Figure 1A). Treatment with compound **1** (25 μM) resulted in a total of 13% early and late apoptotic cells. On the other hand, 48 h incubation with compound **2** (15 μM) yielded 22% Annexin positive and 12% double positive cells. These data suggest that apoptosis is the primary mechanism of compounds **1** and **2** acting against tumor cells.

The activation of caspases was detected in all treatments (Figure 1B), and it increased gradually from Rev-5901 to **1** and **2**, which was demonstrated as an increase in fluorescence shift.

Autophagy was monitored by flow cytometry analysis of acridine orange-stained cells, as the development of acidic autophagosomes is considered to be a hallmark of autophagy [28]. The intensity of fluorescence was significantly increased in colon carcinoma cells treated with **1** (25 μM) and **2** (15 μM) (Figure 1C). On the other hand, Rev-5901 induced a fluorescence shift to the left, suggesting that the number of acridine orange positive cells is lower than in the control sample. This implied that the mode of action of the original inhibitor differs from the mode of action of carborane analogs. The role of autophagy was further investigated by using chloroquine, which is a specific inhibitor of autophagy. This substance acts by changing the lysosomal pH, thereby inhibiting autophagic degradation in the lysosomes [28]. The results of crystal violet assay (Figure 1D) verified that the viability of cells treated with **1** and **2** was significantly reduced in the presence of the autophagic inhibitor, thereby confirming the cytoprotective role of autophagy in CT26CL25 cells.

Furthermore, the Annexin V/propidium iodide staining confirmed the findings of the viability assay (Figure 2). There was a striking increase in a number of late apoptotic cells when treatment with 1 and 2 was combined with chloroquine. In the control sample, we detected 1.17% of late apoptotic cells, and in the chloroquine treated sample, we detected 1.84%. In the combined treatment, 1+ chloroquine, and 2+ chloroquine, values for late apoptotic cells were 77.76% and 71.52%, respectively.

Red fluorescence detected on the FL3 channel was used as positive control for the effect of chloroquine, and it confirmed an inhibition of autophagy.

### 2.2. Intracellular Signaling of **1** Activates MAPK, While **2** Activates the PI3K/Akt Pathway

We investigated two major signaling pathways involved in tumor cell survival and proliferation: the PI3K/Akt pathway and MAPK pathway. Our data show that there is a reciprocal activation of these two pathways (Figure 3). Compound **1** transiently and significantly activates Erk phosphorylation compared to compound **2**. On the other hand, compound **2** rapidly upregulates Akt phosphorylation after 16 h of treatment, while in cells treated with **1**, Akt is downregulated.

Having in mind that apoptotic cell death was detected in colon carcinoma cells treated with **1** and **2**, we tested the expression of pro- and anti-apoptotic proteins, i.e., Bax and Bcl-XL, respectively (Figure 3). There is an early upregulation of Bax detected in cells treated with compound **1** (4 and 6 h), and there is a late discrete upregulation of Bax detected in cells treated with **2** (16 and 24 h). Accordingly, we have detected a transient upregulation of anti-apoptotic Bcl-XL upon treatment with **1** (16 h) and a steady increase in Bcl-XL expression in cells treated with **2**.

### 2.3. Compound **2** Has an In Vivo Potency for Tumor Reduction

In order to confirm the findings described above, we performed an in vivo experiment. Tumors were induced by the subcutaneous inoculation of CT26CL25 tumor cells into Balb/c mice, which were treated with i.p. injections of either Rev-5901, **1** or **2**, in a dosage of 30 mg/kg. At the time of autopsy, the group of animals treated with **2** had significantly lower mean tumor volume compared to all other groups (Figure 4). Surprisingly, treatment with compound **1** did not result in a tumor reduction; on the contrary, this compound potentiated the tumor growth in vivo. The testing of the acute toxicity of compound **2** showed that no mice died of the dose employed. No changes were observed in the mice behavior, food and water intake, as well as in urine parameters of these mice compared to the control group. No macroscopic alterations were observed in the kidneys, lung, spleen or heart in any of the mice.

## 3. Discussion

A link between inflammation and cancer is well established. Pro-inflammatory mediators such as leukotrienes and prostaglandins may promote tumor cell proliferation, growth and metastasis [29]. The involvement of anti-inflammatory agents, such as 5-lipoxygenase inhibitors, can help to suppress cancer growth [30]. Several studies are verifying the pro-apoptotic effect of 5-LO inhibitors in different cancer types. Zileuton, a redox inhibitor, induces apoptosis in pancreatic cancer cells [31] and in cervical cancer (zileuton inhibits arachidonate-5-lipoxygenase to exert antitumor effects in preclinical cervical cancer models). In esophageal cancer, the inhibitors AA861 and Rev-5901 induce a time- and dose-dependent reduction in cell viability [32]. In human bladder tumor cell line T24, treatment with the general lipoxygenase inhibitor NDGA resulted in a cellular shrinkage, chromatin condensation, and the appearance of apoptotic bodies [11]. In colon carcinoma, there is a strong correlation between 5-LO expression and increased polyp size as well as higher tumor grade, suggesting a role for this enzyme in colon cancer development [33].

This work provides in vitro and in vivo evidence of the effect of Rev-5901 carborane analogs on colon carcinoma progression. The mechanism of carborane Rev-5901 analogs has been extensively studied by our team [23,34]. In colon carcinoma, the use of naphthalene and quinoline modifications of Rev-5901, namely compounds **1** and **2**, in vitro induced apoptosis, activation of caspases and increased autophagic activity. These data are in concordance with our previous study where the cytotoxic activity of compounds **1** and **2** was investigated on the A375 melanoma cell line [23]. In this work, the cytotoxic activity of Rev-5901 analogs **1** and **2** was confirmed by detecting an activation of Bax and caspases.

There are reports that the genetic ablation or inhibition of lipoxygenases leads to an enhanced autophagy, resulting in autolysosomal targeting of part of the cytoplasm as well as mitochondria and peroxisomes, thus indicating that autophagy is a compensatory mechanism, which acts as a cleaning system, preventing accumulation of damaged cellular components [35].

Our in vivo data demonstrate the opposite effects of **1** and **2** on colon carcinoma growth. Compound **2** showed a significant tumor shrinkage, while, surprisingly, **1** potentiated tumor growth. These different impacts on tumor progression could be a result of the activation of different signaling pathways. Namely, compound **1** transiently activates the phosphorylation of Erk, implying an activation of the MAPK pathway. The study of Aguirre-Ghiso et al. provides evidence that Erk activity plays an important role in determining tumor growth. Increased levels of Erk activity are necessary for the in vivo growth of cancer cells, which was demonstrated on 10 different cancer cell lines [36]. On the other hand, the activation of the PI3K/Akt pathway also enhances tumor growth in colorectal cancer, as shown in several studies [37,38,39,40]. However, in our tumor model, Akt proves to be beneficial for tumor stagnation. Literature data imply that the activation of Akt does not result solely in tumor development and progression, but it can also play a role in ubiquitination and autophagy [41,42]. Indeed, our results provide evidence for a more intensive autophagy present in cells treated with **2**. A novel study of Chi et al. indicates that the downregulation of 5-Lipoxygenase via the PI3K/Akt signaling pathway results in the reduced proliferation and migration of tumor cells, confirming, once again, the importance of this enzyme in tumor treatment [43]. Nevertheless, one should have in mind that the inhibition of solely 5-LO could result in the intracellular accumulation of prostaglandins [44], raising a need for more detailed investigation of pharmacological effects in future research.

## 4. Materials and Methods

### 4.1. Reagents and Cells

Compounds **1** and **2** were prepared as described [23]. Fetal calf serum (FCS), RPMI-1640 medium, phosphate buffer saline (PBS), dimethyl sulfoxide (DMSO), and chloroquine were from Sigma (St. Louis, MO, USA). The CT26CL25 cell line was purchased from American Type Culture Collection (Rockville, MD, USA). Cells were routinely maintained in HEPES-buffered RPMI-1640 medium supplemented with 10% FCS with 2 mM L-glutamine, 0.01% sodium pyruvate, 100 U/mL penicillin and 100 μg/mL streptomycin at 37 °C in a humidified atmosphere with 5% CO_2_.

### 4.2. Animals

Six to eight-week-old Balb/c male were kept under standard laboratory conditions (nonspecific pathogen-free) with free access to food and water. Animal studies were performed in accordance with EU guidelines and approved by the local Institutional Animal Care and Use Committee (IACUC), approval nr. 03-09/16.

### 4.3. Annexin V/Propidium Iodide Staining

CT26CL25 cells were stained with 1 mg/mL Annexin V-FITC (BD Pharmingen, San Diego, CA, USA) and 1 mg/mL propidium iodide (Sigma, St. Louis, MO, USA) and analyzed using flow cytometry, using the Cy Flow Space (Partec, Munster, Germany).

### 4.4. Activation of Caspases

Cells were incubated with Apostat, a pan-caspase inhibitor (R&D Systems, Minneapolis, MN, USA) in a final concentration of 0.5 µg/mL, for 30 min at 37 °C. After washing with PBS, cells were resuspended and analyzed by flow cytometry.

### 4.5. Detection of Autophagy

Cells were stained with acridine orange (Lab Modena, Paris, France) in a final concentration of 1 μg/mL and incubated for 15 min at 37 °C. Acridine orange was removed by washing with PBS. The cells were resuspended in PBS and analyzed on the green and red channels by flow cytometry.

### 4.6. Western Blot

Cell lysates (20 μg) were separated by 12% SDS-PAGE, electroblotted onto PVDF membrane (Merck, Burlington, MA, USA), blocked with 5% BSA in TBS-Tween (50 mM Tris-HCl pH 7.6, 150 mM NaCl, 0.05% Tween 20), and probed with primary antibody overnight at 4 °C. After washing, membranes were incubated with horseradish peroxidase-conjugated secondary antibody (Santa Cruz Biotechnology, Dallas, TX, USA). Immunoreactive bands were identified by the ECL chemiluminescence detection system (GE Healthcare, Buckinghamshire, UK) according to the manufacturer’s instructions. The following primary antibodies were used: phospho-Akt (Cell Signalling Technology, #4058), Akt (Cell Signalling Technology, #9272), phospho-Erk (Cell Signalling Technology, #4370), Erk (Cell Signalling Technology, #4695), Bax (Santa Cruz Biotechnology, sc-493), Bcl-XL (Santa Cruz Biotechnology, sc-7195), and β-actin (Abcam, ab8227).

### 4.7. Induction of Colon Carcinoma and Animal Treatment

Tumors were induced by the subcutaneous implantation of 3 × 10^6^ CT26CL25 cells in the dorsal right lumbosacral region of Balb/c mice. For tumor inoculation, animals were anaesthetized with ketamine (100 mg/kg) and xylazine (10 mg/kg). When tumors were palpable, 10 days after tumor induction, the animals were randomly allocated to four groups, seven animals per group, and treated daily with Rev-5901, **1** and **2** by i.p. injection of 30 mg/kg body weight for 15 consecutive days. The control group was treated with PBS. Tumors were harvested 25 days after tumor implantation, and their volume was calculated by the formula: (length × width^2^) × 0.52. The acute toxicity of compound **2** was tested by injecting intraperitoneally a single dose of 300 mg/kg in Balb/c mice. The control group was treated with the same volume of PBS. The behavior of animals and food and water intake were observed for 18 days. Urine samples were taken on day 9. Animals were sacrificed, and the major abdominal organs were examined.

### 4.8. Statistical Analysis

We used the Statistical Package for the Social Sciences (SPSS, IBM, Armonk, NY, USA) for data analysis. Student *t*-test, Mann–Whitney test, and one-way ANOVA were employed to evaluate the significance between groups. Two-sided *p*-values of less than 0.05 were considered to indicate statistical significance.

## 5. Conclusions

This work evaluates the mode of action of naphthalene and quinoline Rev-5901 carborane analogs **1** and **2**. The effects of these two compounds on the colon carcinoma cell line are different, resulting in a potentiation or inhibition of in vivo tumor growth. Compound **2** stands out as a valuable new agent for colon cancer treatment.

## Data Availability

Not applicable.

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
