# Peer review of "Carborane-Based Analog of Rev-5901 Attenuates Growth of Colon Carcinoma In Vivo"

_molecules, 2022, doi:10.3390/molecules27144503_

Round 1

Reviewer 1 Report

This paper can be of help to researchers who want to study of meta-carborane. Authors described the application of m-carborane containing Rev-5901 derivatives to cancer  cells. The topic is to introduce an interesting field of carborane chemistry, which is thought to be very valuable. The publication in Molecules is thus recommended without revision.

Author Response

Thank you for your opinion.

English language and style are fine/minor spell check required

We have spellchecked the whole manuscript, thank you for the suggestion.

Reviewer 2 Report

In this study, the authors focused on investigating the effects of the Carboran-based analog of Rev-5901 on colon cancer.

The research is appropriately organized to explain the hypothesis. The findings obtained as a result of the study are well organized and clearly presented.

Although the experimental method performed in the in vivo study method is explained in detail, the applications in cell culture studies are not clearly defined. Molecule 1, molecule 2, apostate, etc., are stated to be applied to cells under the title of 'cell culture studies'. Also, information such as the concentrations of the compounds should be added.

The findings are discussed in detail with reference studies in the 'discussion section'. However, this section of the authors may strengthen the discussion section by supporting more recent references on the topic of study.

After this minor edit, I think the work is suitable for publication.

Kind regards.

Author Response

Thank you for your constructive criticism. Please find below point by point reply. 

Although the experimental method performed in the in vivo study method is explained in detail, the applications in cell culture studies are not clearly defined. Molecule 1, molecule 2, apostate, etc., are stated to be applied to cells under the title of 'cell culture studies'. Also, information such as the concentrations of the compounds should be added.

The concentrations of the compounds are added in the text.

The findings are discussed in detail with reference studies in the 'discussion section'. However, this section of the authors may strengthen the discussion section by supporting more recent references on the topic of study.

The discussion section is improved by adding the latest findings in the field.

Reviewer 3 Report

Molecules-1809429-peer-review-v1

The manuscript by Maksimović-Ivanić et al. evaluates the anticancer effect of 5-lipoxygenase inhibitors, named CarbZDNaph (25 mM) and CarbZDChin (15 mM), compares their activities with commercially available inhibitor Rev-5901 (30 mM). The detailed mechanistic study revealed caspase-dependent apoptosis and autophagy. CarbZDChin molecule showed tumor growth suppression, whereas both CabrZDNaph and Rev-5901 lack the same effect. This present manuscript thus fits well into the Molecules readership's interests and may become suitable for publication after the revision. However, the issues need to resolve before the possible publication of these results.

1. Line 19, abstract section: IC50 needs to be defined in the first place before using the standard abbreviation.

2. The standard deviation values in IC50 are missing; authors need to provide these values before final publications (ref: Inorg. Chem. 2018, 57, 22, 14374-14385).

3. Line 90, Results: Please check the IC50 values stated as “25 μМ, 1 μМ, and 30 μМ for 1, 2” ……is it 1 μМ or 15 μМ?

4. Line 135: In AnnexinV/propidium iodide assay (Figure 2), the control panel (left) showed cells are already in the late apoptosis phase, isn't supposed to be viable only as it might interfere with compound treated results.

5. Line 162, Figure 3 Bcl-XL (right side) looks smeared; the authors might need to reconsider this image before final publications.

6. Line 185: Figure 4 (left side) represents images of tumors after 10 days of treatment presented on the scale? It needs to clarify somewhere in the text (Ref: ACS Omega 2018, 3, 8, 9333-9338).

Author Response

Thank you for your constructive criticism. Your comments helped us to improved our manuscript. Please find below point by point reply.

 Line 19, abstract section: IC50 needs to be defined in the first place before using the standard abbreviation.

We defined the IC50 in the abstract.

  1. The standard deviation values in IC50 are missing; authors need to provide these values before final publications (ref: Inorg. Chem. 2018, 57, 22, 14374-14385).

We have cited our previous publication in which these IC50 values are presented in a table, with standard deviations, reference number 23 (Kuhnert et al., ChemMedChem 2019)

  1. Line 90, Results: Please check the IC50 values stated as “25 μМ, 1 μМ, and 30 μМ for 1, 2” ……is it 1 μМ or 15 μМ?

It is a typo, the correct value is 15 uM.

  1. Line 135: In AnnexinV/propidium iodide assay (Figure 2), the control panel (left) showed cells are already in the late apoptosis phase, isn't supposed to be viable only as it might interfere with compound treated results.

Thank you for your observation. We agree with you that viability of controls can interfere with the sensitivity to the treatment, but in this specific case we omitted to include percentage of apoptotic cells which illustrated that the current state of our control was appropriate. We saw a certain population of cells in a late apoptotic state, however, in percentages, their number is negligable, compared to the total number of cells. Precisely,in the control sample, we detected 6.8% of late apoptotic cells, and in the chloroquine treated sample 6.5%. In the combined treatment, 1+ chloroquine, and 2+ chloroquine, values for late apoptotic cells were 79.2% and 76.5%, respectively. Acceptable percentage of apoptotic cells in control sample is less than 10% like in our case. However, according to your suggestion results were regated and dot plots were replaced with the new ones.Please note that for clarity, we added the percentages on the figure. Experiment is repeated three times and the results are highly reproducibile in terms of effectivness and the outcome of all applied treatments.

  1. Line 162, Figure 3 Bcl-XL (right side) looks smeared; the authors might need to reconsider this image before final publications.

Thank you for your observation. We agree with your comment. However, with used antibody, it was impossible to get more clear blot figure. Since we prefer to present minimaly manipulated western blot figures and we have uploaded the original wb files we just sligtly improved the image. We hope that you will it this more suitable for publication.

  1. Line 185: Figure 4 (left side) represents images of tumors after 10 days of treatment presented on the scale? It needs to clarify somewhere in the text (Ref: ACS Omega 2018, 3, 8, 9333-9338).

No, those are the images of tumors after 15 days of treatment.

The figure legend for Figure 4  says: „Compounds 1, 2 and Rev-5901 were injected i.p. for 15 consecutive days starting on day 10 after tumor implantation. Tumors were harvested 25 days after tumor implantation...“

We have corrected the Materials & Methods section, assuming that caused the confusion.